# The Effects of Temperature on the Quality and Storage Stalibity of Sweet Potato (*Ipomoea batatas* L. [Lam]) Grown in Central Europe

**Barbara Krochmal-Marczak** [1,*] , **Barbara Sawicka** [2] , **Barbara Krzysztofik** [1] , **Honorata Danilčenko** [3] **and Elvyra Jariene** [3]

[1]  Department of Plant Production and Food Safety, Carpathian State College in Krosno, 38-400 Krosno, Poland; krzysztofikb@gmail.com

[2]  Department of Plant Production Technology and Commodity Sciences, University of Life Sciences in Lublin, 20-950 Lublin, Poland; barbara.sawicka@up.lublin.pl

[3]  Agriculture and the Institute of Food Sciences, Vytautas Magnus University Agriculture Academy, Akademija, LT-53361 Kaunas District Municipality, Lithuania; honoratad@gmail.com (H.D.); elvyra58@gmail.com (E.J.)

\*  Correspondence: barbara.marczak@kpu.krosno.pl; Tel.: +4813-437-5580

**Abstract:** The research focuses on the effects of temperature on the quality and storage stability of sweet potato (*Ipomoea batatas* L. [Lam]). It is based on the results of a field experiment conducted between 2015–2017 in Żyznów (49°49′ N, 21°50′ E). The experimental factors were: storage temperatures (5° and 15 °C) and sweet potato cultivars ('Carmen Rubin', 'White Triumph', 'Beauregard', 'Satsumo Imo', 'Purple'). Tubers were harvested at BBCH stage 97. Tubers were stored in a climatic chamber with temperature control and fixed ambient humidity for 6 months during the 2015/16–2017/18 seasons. The storage of tubers at 15 °C helped reduce tuber waste and weight losses resulting from germination, transpiration, respiration and rotting. The quality of the stored tubers depended mostly on storage temperature. It was observed that at 15°, the content of dry matter and total sugars was higher than at 5°, whereas the content of starch was lower. The factor determining storage stability was the genetic features of the cultivars under study. The cultivars 'Purple' and 'Satsumo Imo' demonstrated good storage stability. The cultivar 'Carmen Rubin' turned out least suitable for long storage. The experiment results can contribute to developing storage technology of sweet potato tubers cultivated in Central Europe.

**Keywords:** sweet potato; storage; weight loss; starch; total sugar; storage losses; transpiration; respiration; germination

## 1. Introduction

Sweet potato (*Ipomoea batatas* (L.) Lam) is one of the most important edible plants in the world [1–3]. In many countries, it is used as the basic food product due to its high efficiency, abundant nutrients and possibility of growing in various climates [4–7]. Sweet potato can be grown in extreme conditions in a lot of soil types; hence it plays a huge economic role in a number of countries around the globe. Tuber storage and quality preservation are the key elements in the supply chain. Lack of appropriate, experimentally-proven and tested storage method, as well as storing sweet potato tubers in unsuitable conditions are among the most common reasons of spoilage directly after harvest. During long-term storage of sweet potato tubers, biochemical and physiological processes take place resulting in qualitative and quantitative changes [8,9]. In Poland, sweet potato cultivation started recently and it is only grown as seasonal crop, so in order to make it available for consumption all

year round, it should be stored at the right humidity and temperature. Dandago and Gungula [10], Mbah and Eke-Okoro [11], report that in temperate-climate regions, with production limited to the summer season and constant sales, sweet potato tubers can be stored throughout the entire year, provided that some conditions are met regarding temperature and ambient humidity in the storage room. Sweet potato tuber storage begins after tubers are harvested and is related to good production practices and good storage practices [11,12]. Sweet potato should be harvested in rainless weather at a temperature over 5 °C [11–13]. Harvested tubers should be carefully sorted according to size categories. Only healthy, undamaged tubers with mature, suberized skin are suitable for storage. Sweet potato tubers are characterised by very thin, delicate skin that is prone to damage by cutting or scraping, so during harvest, it is recommended to use paper cartons filled with fewer tubers instead of polypropylene bags [14,15]. Sweet potato storage period can be divided into stages with different recommended temperature and humidity; stage one is tuber maturation, stage two – cooling down and stage three – long-term storage [10,16]. In temperate climate, after harvest and prior to storage, sweet potato tubers should be stored at 12–18 °C for approximately 10 days. During that period, a protective layer of suberized cells forms in tubers, warding off microbes and preventing excessive moisture loss during storage [17]. Tuber storage is instrumental for safeguarding constant supplies for the food processing industry for the production of juices, soups and in particular fried products, i.e., crisps and fries. Therefore, preservation of sweet potato tuber quality in storage is necessary both for the processing industry and to avoid high economic losses. The preference is to store sweet potato tubers in a cool store in order to maintain their good quality. However, long-term low-temperature storage results in cold-induced sweetening (CIS), leading to accumulation of sugars and fried food browning [13,18]. Hence, the aim of this research was to determine the effects of storage temperature on storage stability and quality of tubers of several cultivars and as a result to develop the basis for sweet potato tuber storage technology suitable for the Central European cultivation conditions.

## 2. Materials and Methods

The study was based on tuber samples from a field experiment conducted in slightly acidic brown earth [19], between 2015–2017 in Żyznów (49°49′ N 21°50′ E), Poland. The experiment was based on a randomised block design in three replicates. The research objects were sweet potato (*Ipomoea batatas* L. [Lam]) cultivars. The storage experiment focused on two factors. The primary experimental factor was the air temperature in the climatic chamber (5° and 15 °C), and the secondary factors were the cultivars ('Carmen Rubin', 'White Triumph', 'Beauregard', 'Satsumo Imo', 'Purple'). The basic characteristics of all sweet potato cultivars are described in the Table 1.

**Table 1.** Characteristics of the tested cultivars.

| Parameter | Cultivars | | | | |
|---|---|---|---|---|---|
| | 'Carmen Rubin' | 'White Triumph' | 'Beauregard' | 'Satsumo Imo' | 'Purple' |
| **Shape** | ovate | ovate | ovate | oblong | ovate/irregular/oblong |
| **Skin colour** | pink | white | orange | purple | purple |
| **Flesh colour** | orange | white | dark orange | cream | purple |
| **Earliness group** | medium-early | medium-early | early late | medium-late | late |
| **Length of the growing season** | 105–110 | 100–120 | 90–100 | 110–130 | 120–160 |

### 2.1. Collecting Soil Samples

Annually, prior to starting the experiment, 20 primary soil samples were collected from each of them, all making up the bulk sample, with a weight of approx. 0.5 kg [20]. The samples were tested

for: the granulometric composition of the soil, pH in 1 mol KCl.dm$^{-3}$ and the content of total organic carbon (C$_{org}$)–with Tiurin's method, and on that basis, the content of humus in soil was tested [21].

## 2.2. Field Experiment Methodology

The pre-crop in the field experiment was spring barley. Pre-crop harvest was followed by shallow ploughing. In autumn, manure was used in the amount of 25 t·ha$^{-1}$ as well as phosphorus and potassium fertilization in the amount of 34.9 kg P and 99.6 kg K·ha$^{-1}$. The dose size of PK fertilizers was determined based on the amount of these ingredients in soil. In spring, prior to planting, nitrogen fertilization was applied in the amount of 80 kg·N·ha$^{-1}$ mixed with soil with a soil cultivator. The propagating material included healthy rooted cuttings of sweet potato from in vitro propagation. They were planted with $40 \times 75$ cm spacing in mid-May. The size of crop plots was 15 m$^2$. During vegetation, cultivation was carried out in accordance with normal agricultural practice. Mechanical or manual weeding was conducted since row closure. After row closure, the plants shaded the row spacing and there was no need for any weeding processes. In addition, no pathogen protection was applied, because no diseases or pests were found on sweet potatoes. Tubers were harvested at Biologische Bundesanstalt, Bundessortenamt und Chemische Industrie (BBCH) stage 97 [22]. In 2015, harvest took place on 12–13 October, in 2016—on 17–18 October, and in 2017—on 16–17 October.

## 2.3. Collecting Tuber Samples

During harvest, tuber samples were collected for direct post-harvest assessment (50 medium-sized tubers each) and samples for the storage experiment. For the purpose of storage, six 10-kg samples were collected from each field experiment combination. Prior to entering the samples in the climatic chamber, mechanical damages were weighed. Tuber samples of each cultivar underwent maturation process. To this end, sweet potato tuber samples were placed in the climatic chamber (type KBK-100 Wamed, Warsaw, Poland), with air temperature set at 18 °C and humidity at 95% for the first 10 days [16]. Then, the air temperature was gradually decreased to 5° and 15 °C. From the beginning of November until the end of the storage period (end of April), the air temperature and humidity in the chamber were maintained at a constant level, i.e., 5° and 15 °C, whereas relative humidity in the storage chamber was 95%.

## 2.4. Storage Stability Test

The storage experiment was completed at the end of the 3rd decade of April, after 6 months of storage. Upon completion, the following storage stability elements were measured: sprout length, natural tuber weight losses resulting from transpiration and respiration and waste resulting from fungal and bacterial diseases. Each of the affected tubers was cut, the disease was identified and the percentage of storage waste due to tuber rotting was determined.

The beginning of germination and the intensity of sprout growth was measured in tubers stored in openwork cases. The assessment was conducted every 10 days, and it was assumed that the beginning of sprouting was the time when 75% of tubers had sprouts exceeding 3 mm in length. After the storage season, samples were subject to assessment of losses resulting from transpiration, respiration and sprouting, as well as waste due to storage diseases. Natural losses (P$_{po}$) were calculated with the following formula [23]:

$$P_{po} = \frac{A_1 - A_2}{A_1} * 100 \ (\%).$$

(1)

where: A$_1$—sample weight prior to storage (g), A$_2$—sample weight after storage (g).

The value of storage losses (W$_{ch}$) caused by dry, wet and mixed rot was calculated with the following formula [23]:

$$W_{ch} = \frac{A_{ch}}{A_1} 100 (\%).$$

(2)

where: A$_{ch}$—diseased tuber weight (g), A$_1$—sample weight prior to storage (g).

### 2.5. Tuber Quality Test

In each year, following tuber harvest and after 6 months of storage, dry matter, starch and total sugars were measured.

### 2.5.1. Dry Matter Content

Dry matter content in tubers was measured with a heating and weight method. Prior to dry matter measurement, tubers were washed in cold water, drained and dried at room temperature. Analyses were conducted in three replicates for each sample. A glass weighing bottle was first dried in a heating chamber at 105 °C for 2 h, then cooled down to room temperature (approx. 30 min.), and weighed to an accuracy of 0.001 g. Then, ground plant material and fresh fragmented plant material were put in the bottle, which was placed in the heating chamber and dried at 60 °C for approx. 12 h, and then at 105 °C for 2 h. The bottle with the dry matter sample was placed in a desiccator, cooled down and weighed. Drying was considered complete when the difference between two subsequent weightings did not exceed 0.001 g. Dry matter content in the sample was calculated with the following formula:

$$DM = \frac{c - a}{b - a} * 100 (\%) \tag{3}$$

where $a$—bottle weight, $b$—bottle weight with fresh plant material, $c$–bottle weight with plant material after drying at 105 °C [24].

### 2.5.2. Sugar Total Content

Five g of mashed sweet potato were put into a sealed test tube. 5 mL of 80% ethanol was added and well-mixed. The test tube was placed in a water-bath at 80 °C for 15 min. Subsequently, 2.5 mL of 80% ethanol was added, and the mixture was heated for 15 min. Then another 2.5 mL of 80% ethanol was added, and the mixture was heated for another 15 min. The mixture was filtered using Whatman No. 4 filtering paper. The filtrate was collected. Approximately 5 mL of filtrate was filtered through a 0.45 µm membrane filter before injecting it into the high-performance liquid chromatograph (HPLC). The sugar content determination was based on a modified method provided by Picha [25]. Sugars were separated using the HPLC (Pump model No. LC1150, GBC Scientific Equipment, Dandenong, Australia) with a reverse-phase C-18 column (Model No. LiChrolut RP-18, Merck, Darmstadt, Germany). A refractive index detector (Model No. LC1246K, GBC Scientific Equipment, Hampshire, USA) was used to analyse the signals. The heating oven temperature was 45 °C. The mobile phase was a mixture of degassed HPLC-grade acetonitrile (80%) and distilled water (20%). The flow rate was 1 mL/min. Four sugar standards, including glucose, fructose, sucrose, and maltose (Sigma Chemical Co, St. Louis, MO, USA), were prepared individually. Each sugar standard was prepared by dissolving 0.01 g of sugar into 1 mL of distilled water [26].

### 2.5.3. Starch Content

The total starch content of sweet potatoes was determined by using the method of International Starch Institute–Denmark described by Woolfe [27]. The residue left in the test tube in times of sugar estimation was taken in a test tube, dried at 70 °C. Water was added in the sample (5 mL), cooled in ice bath and 6.5 mL perchloric acid was added. Then centrifuged and clear supernatant was separated. This process was repeated for 3 times. 0.1 mL sample solution was taken in a test tube, perchloric acid and anthrone reagent (2 mL) was added to the sample and samples were measured for 630 nm using a spectrophotometer (model 6850, Jenway, Staffordshire, UK) [28].

### 2.6. Weather Conditions

The humidity and temperature conditions during sweet potato vegetation were described with the use of the hydrothermal coefficient of Selyaninov, as illustrated in Table 2.

**Table 2.** Hydrothermal coefficient of Selyaninov during sweet potato vegetation period in 2015–2017 according to the COBORU meteorological station in Dukla.

| Year | Month | Mean Rainfall [mm] | Temperature [°C] | | | Mean Temperature [°C] | Hydrothermal Coefficient of Selyaninov |
| | | | Decade | | | | |
| | | | I | II | III | | |
|------|-------|------|------|------|------|------|------|
| 2015 | May | 70.4 | 5.1 | 7.2 | 7.5 | 19.8 | 3.6 |
| | June | 60.2 | 14.2 | 12.3 | 8.9 | 35.4 | 1.7 |
| | July | 178.1 | 14.1 | 14.2 | 15.5 | 43.8 | 4.1 |
| | August | 107.2 | 17.2 | 15.4 | 23.2 | 55.8 | 1.9 |
| | September | 63.2 | 21.1 | 24.1 | 21.1 | 66.3 | 1.0 |
| | October | 95.3 | 15.2 | 10.2 | 18.4 | 43.8 | 2.2 |
| Average | | 957 | 14.5 | 13.9 | 15.8 | 44.1 | 2.4 |
| 2016 | May | 78.2 | 8.2 | 8.6 | 10.4 | 27.2 | 2.6 |
| | June | 91.2 | 14.1 | 9.2 | 8.6 | 31.9 | 2.0 |
| | July | 88.1 | 20.3 | 20.1 | 19.9 | 60.3 | 1.5 |
| | August | 172 | 20.1 | 19.6 | 15.0 | 60.3 | 2.9 |
| | September | 110.3 | 18.7 | 19.3 | 16.1 | 54.1 | 2.0 |
| | October | 9.4 | 17.6 | 16.9 | 14.1 | 48.6 | 0.2 |
| Average | | 291.2 | 16.5 | 14.0 | 14.0 | 46.9 | 1.9 |
| 2017 | May | 70.4 | 8.6 | 8.5 | 9.8 | 26.9 | 2.6 |
| | June | 180.2 | 17.6 | 16.5 | 15.9 | 50 | 3.6 |
| | July | 18.2 | 20.3 | 21.6 | 22.0 | 63.9 | 0.3 |
| | August | 120.1 | 19.7 | 18.6 | 19.9 | 58.2 | 2.1 |
| | September | 15.1 | 13.5 | 14.2 | 10.1 | 37.8 | 0.4 |
| | October | 20.5 | 15.6 | 14.2 | 14.3 | 44.1 | 0.5 |
| Average | | 236.3 | 15.9 | 15.6 | 14.6 | 46.8 | 1.6 |

Source: own study according to data from the COBORU meteorological station at SDOO in Dukla. The following ranges of values $p_{0.05}$ for the Selyaninov coefficient were assumed: extremely dry k ≤ 0.4; very dry 0.4 < k ≤ 0.7; dry 0.7 <k ≤ 1.0; quite dry 1.0 < k ≤ 1.3; optimal 1.3 < k ≤ 1.6; quite damp 1.6 < k ≤ 2.0; wet 2.0 < k ≤ 2.5; very wet 2.5 < k ≤ 3.0; extremely humid k > 3.0.

The years 2016–2017 were considered wet, whereas 2015 was extremely humid, which is reflected in the values of the hydrothermal coefficient of Selyaninov (Table 2). However, significant variation of the hydrothermal coefficient was observed between individual months of the vegetation period. In 2015, all vegetation months were wet, with an extremely humid May and July and a dry September. In 2016, almost all months, except for October, were wet or very wet. In 2017, July and September were extremely dry, with a dry October, whereas all the other months were wet or very wet, with an extremely humid June.

*2.7. Statistical Analysis*

Statistical analyses were based on two-factor analysis of variance and Tukey's multiple tests, with the assumed significance level $p_{0.05}$. The models of analysis of variance with the main effects of the factors studied and their interactions were used. The detailed analysis only dealt with the main effects. The calculations were made with the SAS Enterprise 4.2 software [29]. Tukey's multiple comparison tests enabled detailed comparative analyses of averages, by isolating statistically homogeneous medium groups (homogeneous groups) and determining the so-called least significant mean differences (LSD), which in Tukey's tests are marked by LSD (Tukey's Honest Significant Difference). The calculated *p*-values determine the significance and magnitude of the impact of the studied factors on the

differentiation of the results of the analysed variables, by comparing them with the most frequently accepted levels of alpha significance (0.05, 0.01). In case of detailed analyses based on Tukey's multiple tests, the significance level $p = 0.05$ was assumed. Letter indicators at averages determine the so-called homogeneous groups (statistically homogeneous). The occurrence of the same letter pointer at averages (at least one) means that there is no (no) statistically significant difference between them. The sizes of LSD play an auxiliary role, allowing to quantify the differences between means in a quantitative way. The elaboration of the results concerning the data expressed as percentages was performed on the transformed values. The change of percentages to angular degrees (Bliss transformation) was made according to the following formula:

$$Z = arcsin \sqrt{y} \tag{4}$$

where $y$–% of tubers with or without a given value (e.g., tuber is rotten or not). Such transformations are used for data with a binomial distribution expressed as a percentage, most often taking values in the range of 0–20% or 80–100%. In our case, we deal with a binomial distribution, so the transformations were applied before the analysis of variance was performed [30].

## 3. Results

### 3.1. Soil Testing

The experiment was conducted in soil consisting of flysch sediments, referred to as Carpathian loess or carbonate-free loess-like soil. Classification of soils in this region: autogenic–luvisols–lessive–pseudogley [19]. Sand fraction amounted to 37.16%, dust fraction–56.70% and clay–6.14% (Table 3). Such fraction share is an equivalent of clayey dust [19]. In terms of agricultural suitability, these soils are characterised by good rye complex, valuation class IVb, slightly acidic pH (5.69 pH in 1n KCl) (Table 4). In terms of agronomic weight, this soil belongs to medium-textured lessive soil [31].

**Table 3.** The granulometric composition of soil in %.

| Year | Composition Content of the Granulometric Fractions (%) | | | | | | Soil Classification |
|---|---|---|---|---|---|---|---|
| | **2.0–1.0** | **1.0–0.5** | **0.5–0.05** | **0.05–0.005** | **0.005–0.002** | **<0.002** | |
| **2015** | 0.00 | 2.96 | 34.22 | 48.08 | 8.59 | 6.15 | Pyg/SiL * |
| **2016** | 0.00 | 2.84 | 34.30 | 48.02 | 8.71 | 6.13 | Pyg/SiL * |
| **2017** | 0.00 | 2.76 | 34.26 | 48.21 | 8.62 | 6.15 | Pyg/SiL * |
| **Average** | 0.00 | 2.85 | 34.26 | 48.10 | 8.64 | 6.14 | |

\* pyg–SiL–sility loamSource: own study based on the results of the Regional Chemical and Agricultural Station in Rzeszów.

The concentration of assimilable phosphorus and potassium in soil was on a medium level, with a very high content of magnesium, and also a medium level of copper, manganese, iron and zinc. The average content of humus in the topsoil was 2.72 g kg$^{-1}$, with a slight acidity of soil (Table 4) [31].

**Table 4.** Physico-chemical characteristics of the soil before establishing the experiment in 2015–2017.

| Years | Macronutrients (mg·100 g$^{-1}$ of Soil) | | | CaCO$_3$ (g kg$^{-1}$) | Humus (g kg$^{-1}$) | pH (KCL) | Micronutrients (mg·kg$^{-1}$ of Soil) | | | |
|---|---|---|---|---|---|---|---|---|---|---|
| | **P$_2$0$_5$** | **K$_2$0** | **Mg** | | | | **Cu** | **Mn** | **Zn** | **Fe** |
| **2015** | 12.4 | 20.1 | 19.8 | 0.02 | 2.69 | 5.56 | 5.61 | 177.1 | 14.4 | 1581 |
| **2016** | 12.2 | 20.1 | 19.3 | 0.03 | 2.73 | 5.73 | 5.70 | 172.9 | 14.5 | 1572 |
| **2017** | 12.3 | 20.4 | 19.5 | 0.02 | 2.75 | 5.72 | 5.61 | 177.0 | 14.4 | 1569 |
| **Average** | 12.3 | 20.2 | 19.5 | 0.02 | 2.72 | - | 5.64 | 175.7 | 14.4 | 1574 |

Source: data was compiled on the basis of the results obtained by the District Chemical and Agricultural Station in Rzeszów (2015–2017).

## 3.2. Tuber Weight Losses and Waste during Storage

Average natural tuber losses and waste during 6 months of storage were as follows: natural sprouting losses amounted to just 0.41%, natural transpiration and respiration losses–5.61%, rotting–26.56% waste of tuber weight. In total, tuber weight losses and waste during 6 months of storage amounted to 32.59%, depending on temperature–2.17–53.01%, depending on cultivar–27.68–40.15% and depending on storage season–31.98–33.29% (Table 5).

**Table 5.** Sweet potato tuber natural losses and rotting waste depending on storage temperature, cultivar and storage season (%).

| Experimental Factors | | Natural Sprouting Losses | Natural Transpiration and Respiration Losses | Tuber Rotting Waste | Total |
|---|---|---|---|---|---|
| **Storage temperature** | 5 °C | 0.29 ± 0.07 [b] | 4.28 ± 0.72 [a] | 48.44 ± 8.38 [a] | 53.01 |
| | 15 °C | 0.54 ± 0.12 [a] | 6.94 ± 0.63 [b] | 4.69 ± 1.19 [b] | 12.17 |
| **Cultivars** | 'Carmen Rubin' | 0.51 ± 0.19 [a] | 6.78 ± 1.26 [a] | 32.86 ± 28.62 [a] | 40.15 |
| | 'White Triumph' | 0.37 ± 0.16 [b] | 5.29 ± 0.75 [b] | 29.37 ± 26.05 [a] | 35.03 |
| | 'Beauregard' | 0.43 ± 0.06 [b] | 5.75 ± 0.37 [a] | 24.29 ± 18.65 [b] | 30.47 |
| | 'Satsumo Imo' | 0.40 ± 0.08 [b] | 5.10 ± 0.77 [c] | 24.11 ± 19.77 [b] | 29.61 |
| | 'Purple' | 0.36 ± 0.09 [c] | 5.15 ± 0.45 [c] | 22.17 ± 19.98 [c] | 27.68 |
| **Years** | 2015/2016 | 0.38 ± 0.13 [b] | 5.19 ± 0.95 [b] | 27.72 ± 24.24 [a] | 33.29 |
| | 2016/2017 | 0.42 ± 0.13 [a] | 5.58 ± 0.99 [a] | 26.51 ± 23.11 [b] | 32.51 |
| | 2017/2018 | 0.45 ± 0.17 [a] | 6.06 ± 0.92 [b] | 25.47 ± 21.66 [c] | 31.98 |
| **Average** | | 0.41 | 5.61 | 26.56 | 32.59 |

Letter indicators (a, b, c, etc.) next to the averages refer to the so-called homogeneous groups (statistically homogeneous). The occurrence of the same letter indicator next to averages (at least one) means that there is no statistically significant difference in $p_{0.05}$ between them.

## 3.3. Tuber Sprouting

The beginning of tuber sprouting in the experiment depended on storage temperature and cultivar. Genetic features of the cultivars had the largest influence on the beginning of sprouting. The earliest germination was observed in the 'Carmen Rubin' cultivar, and the latest in 'Purple'. Cultivars stored at 5 °C began to sprout in the third decade of February, whereas cultivars stored at 15 °C–already in the first decade of January. The beginning of sprouting was also influenced by the weather conditions in the experiment years. Sweet potato tubers grown between 2015–2016, characterised by high rainfall, began to sprout at a later time, whereas in 2017, characterised by drought in the final period before harvest, began to sprout at the earliest time (Table 6).

**Table 6.** The beginning of sweet potato tuber germination during storage depending on cultivar, temperature and storage period.

| Experimental Factors | | Storage Seasons | | | | | |
|---|---|---|---|---|---|---|---|
| | | **2015/2016** | | **2016/2017** | | **2017/2018** | |
| | | **Day** | **Month** | **Day** | **Month** | **Day** | **Month** |
| **Storage Temperature** | 5 °C | 26th | February | 20th | February | 14th | February |
| | 15 °C | 16th | January | 16th | January | 6th | January |
| **Cultivars** | 'Carmen Rubin' | 5th | February | 7th | February | 3rd | February |
| | 'White Triumph' | 4th | March | 6th | March | 2nd | March |
| | 'Beauregard' | 9th | February | 12th | February | 10th | February |
| | 'Satsumo Imo' | 15th | February | 17th | February | 12th | February |
| | 'Purple' | 18th | April | 22nd | April | 10th | April |

Sprout length after the storage period was significantly influenced by storage temperature. At 15 °C, sweet potato tubers developed sprouts almost 3 times longer than tubers stored at 5 °C. On average, the longest sprouts during storage were developed in 'Carmen Rubin', and the shortest in 'Purple'. The 'White Triumph' and 'Purple', as well as 'Beauregard', 'Satsumo Imo' and 'Carmen Rubin' cultivars turned out to be homogeneous in terms of this characteristic. The longest sprouts were developed in tubers in the 2017/2018 season, when in the year prior to storage, the plants were exposed to excessive rainfall (water stress), and the shortest in the 2016/2017 season, when in the year prior to storage, the weather was changing and September was characterised by a significant rainfall shortage. However, the length of sprouts in the first and last storage season did not differ significantly. The cultivars under study responded differently to storage conditions in the 2015/2016 and 2016/2017 season. In both seasons, the longest sprouts were developed in the 'Carmen Rubin' cultivar and the shortest in 'Purple', whereas in the season 2017/2018, preceded by a very wet vegetation period, the cultivars did not differ significantly in terms of this characteristic (Table 7).

**Table 7.** Sprout length in *Ipomoea batatas* tubers after 6 months of storage (mm).

| Experimental Factors | | Storage Seasons | | | Average |
|---|---|---|---|---|---|
| | | **2015/2016** | **2016/2017** | **2017/2018** | |
| **Storage Temperature** | 5 °C | 3.27 ± 0.88 [b] | 3.13 ± 1.10 [b] | 3.60 ± 0.83 [b] | 3.33 [b] |
| | 15 °C | 11.07 ± 1.98 [a] | 10.47 ± 1.96 [a] | 12.13 ± 1.73 [a] | 11.22 [a] |
| **Cultivars** | 'Carmen Rubin' | 9.00 ± 5.14 [a] | 8.50 ± 5.68 [a] | 8.83 ± 6.05 [a] | 8.78 [a] |
| | 'White Triumph' | 6.00 ± 4.05 [b] | 6.50 ± 4.59 [a] | 7.83 ± 5.31 [a] | 6.78 [b] |
| | 'Beauregard' | 7.33 ± 3.67 [a] | 5.83 ± 2.79 [ab] | 6.83 ± 3.19 [a] | 6.67 [b] |
| | 'Satsumo Imo' | 7.67 ± 5.50 [a] | 7.50 ± 3.01 [a] | 8.67 ± 4.41 [a] | 7.94 [a] |
| | 'Purple' | 5.83 ± 3.13 [b] | 5.67 ± 3.87 [b] | 7.17 ± 4.62 [a] | 6.22 [b] |
| **Average** | | 7.17 [a] | 6.80 [b] | 7.87 [a] | 7.28 |

Letter indicators (a, b, c, etc.) next to the averages refer to the so-called statistically homogeneous groups. The same letter indicator next to averages (at least one) means that there is no statistically significant difference in $p_{0.05}$ between them.

Tuber weight losses due to sprouting were significantly influenced by storage temperature. Such losses were almost twice as high in the case of storage at 15 °C, as compared to storage at 5 °C. Genetic features of the cultivars under study also had a significant influence on the amount of natural losses. The cultivar most prone to germination was 'Carmen Rubin', and the least–'Purple'; whereas the cultivars homogeneous in terms of this characteristic were: 'Beauregard', 'White Triumph' and 'Satsumo Imo'. Storage seasons also differentiated the sprouting tendency of tubers. The greatest sprouting weight losses in tubers were observed in the 2017/2018 season, preceded by high rainfalls in the vegetation period but also drought prior to harvest, and the least sprouting weight losses were recorded in the 2015/2016 season, preceded by a wet and cool vegetation period. The response of cultivars to vegetation conditions varied in the storage seasons. In the seasons 2015/2016 and 2017/2018, the greatest natural losses were recorded in the 'Carmen Rubin' cultivar, and the least in 'White Triumph' in the first season, and in 'Purple' in the last storage season (Table 8).

**Table 8.** Natural sprouting losses depending on cultivar and storage temperature (mm) (Average from seasons 2015/2016, 2016/2017, 2017/2018).

| Experimental Factors | | Storage Seasons | | | Average |
|---|---|---|---|---|---|
| | | 2015/2016 | 2016/2017 | 2017/2018 | |
| Storage Temperature | 5 °C | 0.26 ± 0.03 [b] | 0.30 ± 0.04 [b] | 0.31 ± 0.04 [b] | 0.29 [b] |
| | 15 °C | 0.49 ± 0.10 [a] | 0.54 ± 0.10 [a] | 0.59 ± 0.14 [a] | 0.54 [a] |
| Cultivars | 'Carmen Rubin' | 0.45 ± 0.20 [a] | 0.50 ± 0.23 [a] | 0.59 ± 0.27 [a] | 0.51 [a] |
| | 'White Triumph' | 0.28 ± 0.08 [b] | 0.39 ± 0.17 [a] | 0.44 ± 0.20 [a] | 0.37 [b] |
| | 'Beauregard' | 0.41 ± 0.14 [a] | 0.42 ± 0.11 [a] | 0.45 ± 0.10 [a] | 0.43 [b] |
| | 'Satsumo Imo' | 0.39 ± 0.10 [a] | 0.41 ± 0.07 [a] | 0.41 ± 0.08 [b] | 0.40 [b] |
| | 'Purple' | 0.35 ± 0.12 [a] | 0.38 ± 0.08 [a] | 0.36 ± 0.10 [b] | 0.36 [c] |
| Average | | 0.38 [b] | 0.42 [a] | 0.45 [a] | 0.42 |

Letter indicators (a, b, c, etc.) next to the averages refer to the so-called statistically homogeneous groups. The same letter indicator next to averages (at least one) means that there is no statistically significant difference in $p_{0.05}$ between them.

### 3.4. Transpiration and Respiration Weight Losses

Natural losses due to transpiration and vaporisation during 6 months of storage in individual storage seasons were different and ranged from 4.03 to 7.59%. The factor determining this characteristic the most was storage temperature. At 15 °C, transpiration and respiration resulted in greater losses than at 5 °C (Table 9).

**Table 9.** Natural transpiration and respiration losses depending on storage temperature, cultivar and storage season (%).

| Experimental Factors | | Storage Seasons | | | Average |
|---|---|---|---|---|---|
| | | 2015/2016 | 2016/2017 | 2017/2018 | |
| Storage Temperature | 5 °C | 4.03 ± 0.84 [b] | 4.27 ± 0.61 [b] | 4.53 ± 0.70 [b] | 4.28 [b] |
| | 15 °C | 6.36 ± 1.21 [a] | 6.88 ± 1.06 [a] | 7.59 ± 0.82 [a] | 6.94 [a] |
| Cultivars | 'Carmen Rubin' | 6.63 ± 1.64 [a] | 6.61 ± 1.95 [a] | 7.09 ± 1.92 [a] | 6.78 [a] |
| | 'White Triumph' | 5.20 ± 0.57 [a] | 5.18 ± 1.12 [a] | 5.48 ± 1.27 [b] | 5.29 [b] |
| | 'Beauregard' | 5.55 ± 1.74 [a] | 5.73 ± 2.19 [a] | 5.96 ± 2.11 [a] | 5.75 [b] |
| | 'Satsumo Imo' | 4.61 ± 1.62 [a] | 4.98 ± 1.53 [a] | 5.70 ± 2.23 [a] | 5.10 [b] |
| | 'Purple' | 3.98 ± 0.90 [b] | 5.38 ± 0.40 [a] | 6.09 ± 0.83 [a] | 5.15 [b] |
| Average | | 5.19 | 5.58 | 6.06 | 5.61 |

Letter indicators (a, b, c, etc.) next to the averages refer to the so-called statistically homogeneous groups. The occurrence of the same letter indicator next to averages (at least one) means that there is no statistically significant difference in $p_{0.05}$ between them.

Genetic features of the cultivars had a significant influence on the amount of losses during tuber storage. The greatest transpiration and respiration losses were found in the 'Carmen Rubin' cultivar, and the least in 'Purple'; whereas 'White Triumph', 'Beauregard', 'Satsumo Imo' and 'Purple' belonged to the same homogeneous group in terms of this characteristic. The greatest natural losses of tuber weight due to transpiration and respiration occurred in the 2017/2018 storage period, with a warm and very wet year preceding tuber storage, and the least in the 2015/2016 season; whereas the losses were homogeneous as in the 2016/2017 season. The response of cultivars to storage conditions varied in seasons 2015/2016 and 2017/2018. In both seasons, the greatest transpiration and respiration losses were observed in the 'Carmen Rubin' cultivar, and the least in the 2015/2016 were found in 'Purple', and in the 2017/2018–in 'White Triumph' (Table 9).

The interaction between cultivars and storage seasons points to discrepancies in transpiration and respiration weight losses in *Ipomoea batatas* L. [Lam] tubers. Only the 'Purple' cultivar responded with a significant increase in transpiration and respiration weight loss in the 2017/2018 season as compared to the 2015/2016 storage season. In the case of the remaining cultivars, the differences were insignificant (Figure 1).

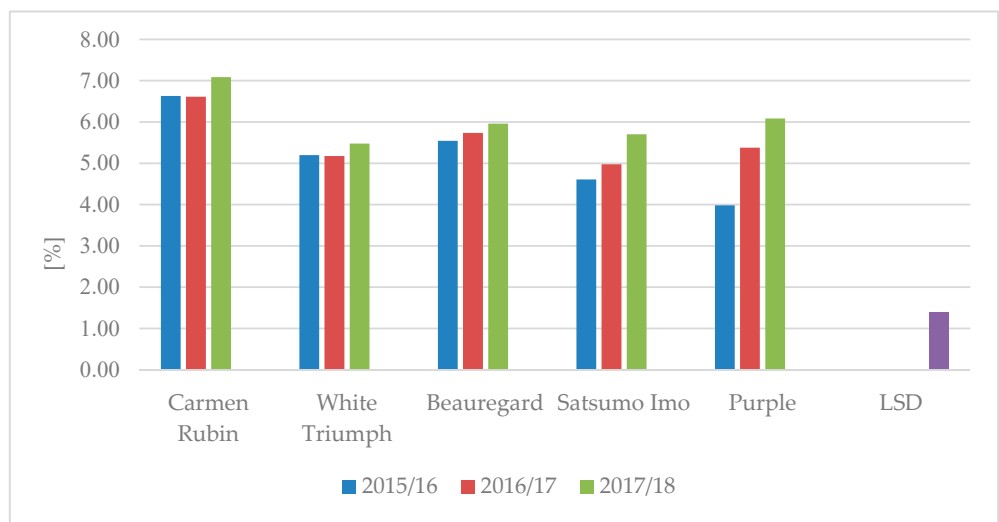

**Figure 1.** The influence of cultivar and storage season on transpiration and respiration losses.

### 3.5. Tuber Waste Due to Rotting

The greatest storage waste during 6 months of storage was due to tuber rotting. The factor differentiating this characteristic the most was storage temperature. Tubers stored at 5 °C were several times more prone to rotting that tubers stored at 15 °C. Tubers stored at lower temperatures were more susceptible to rot despite lower water loss. The cultivar most prone to tuber rotting was 'Carmen Rubin', and the least–'Purple', whereas 'Beauregard' and 'Satsumo Imo' turned out to be homogeneous in terms of this characteristic.

In storage seasons 2015/2016 and 2016/2017, the cultivar most resistant to tuber rotting was Purple, whereas the remaining cultivars turned out to be homogeneous in terms of this characteristic. In season 2017/2018, all cultivars were homogeneous in terms of this characteristic (Table 10).

**Table 10.** Waste due to tuber rotting depending on storage temperature, cultivar and storage season (%).

| Experimental Factors | | Storage Seasons | | | Average |
|---|---|---|---|---|---|
| | | **2015/2016** | **2016/2017** | **2017/2018** | |
| **Storage Temperature** | 5 °C | 50.63 ± 9.53 [a] | 48.30 ± 9.31 a | 46.38 ± 5.80 [a] | 48.44 [a] |
| | 15 °C | 4.80 ± 1.06 [b] | 4.71 ± 1.36 [b] | 4.55 ± 1.21 [b] | 4.69 [b] |
| **Cultivars** | 'Carmen Rubin' | 34.79 ± 32.50 [a] | 33.31 ± 30.46 [a] | 30.56 ± 28.10 [a] | 32.86 [a] |
| | 'White Triumph' | 31.37 ± 29.84 [a] | 30.95 ± 28.83 [a] | 25.80 ± 23.71 [a] | 29.37 [a] |
| | 'Beauregard' | 25.49 ± 21.19 [a] | 23.84 ± 19.34 [a] | 23.53 ± 18.90 [a] | 24.29 [b] |
| | 'Satsumo Imo' | 24.44 ± 20.94 [a] | 23.02 ± 19.85 [a] | 24.88 ± 22.23 [a] | 24.11 [b] |
| | 'Purple' | 22.52 ± 21.13 [b] | 21.42 ± 20.91 [b] | 22.58 ± 21.75 [a] | 22.17 [b] |
| **Average** | | 27.72 | 26.51 | 25.47 | 26.56 |

Letter indicators (a, b, c, etc.) next to the averages refer to the so-called statistically homogeneous groups. The same letter indicator next to averages (at least one) means that there is no statistically significant difference in $p_{0.05}$ between them.

The vegetation period in 2015 was characterised by high rainfall, in particular in July, which is a month of intensive tuber crop increase. Weather conditions in 2015 could have contributed to the spread of storage diseases. Only in the 2017/2018 season did sweet potato cultivars manifest no significant differences in terms of resistance to fungal and bacterial diseases. In the remaining storage seasons, the cultivar most resistant to storage diseases was 'Purple', and the least–'Carmen Rubin' (Table 10).

### 3.6. Tuber Quality after 6 Months of Storage

An analysis of selected chemical components of sweet potato tubers directly after harvest revealed significant differentiation depending on cultivar and weather conditions during sweet potato vegetation (Table 11).

**Table 11.** The content of dry matter, starch and total sugars in sweet potato tubers after harvest.

| Experimental Factors | | Dry Matter (g kg⁻¹ DM) | Starch (g kg⁻¹ FM) | Total Sugar (g kg⁻¹ FM) |
|---|---|---|---|---|
| **Cultivars** | 'Carmen Rubin' | 21.59 ± 0.91 [c] | 15.54 ± 1.51 [c] | 7.97 ± 0.65 [a] |
| | 'White Triumph' | 29.32 ± 0.72 [a] | 21.11 ± 0.49 [a] | 3.30 ± 0.35 [d] |
| | 'Beauregard' | 23.76 ± 0.49 [b] | 17.11 ± 0.25 [b] | 6.04 ± 0.20 [b] |
| | 'Satsumo Imo' | 29.92 ± 0.72 [a] | 21.54 ± 0.39 [a] | 5.08 ± 1.08 [c] |
| | 'Purple' | 28.47 ± 1.13 [a] | 20.50 ± 0.38 [a] | 3.97 ± 0.29 [d] |
| **Years** | 2015 | 25.82 ± 3.47 [b] | 18.59 ± 0.28 [b] | 5.10 ± 1.89 [a] |
| | 2016 | 26.58 ± 3.45 [a] | 19.14 ± 1.46 [a] | 4.86 ± 1.58 [b] |
| | 2017 | 27.43 ± 3.45 [a] | 19.75 ± 1.72 [a] | 5.86 ± 1.75 [a] |
| **Average** | | 26.61 | 19.16 | 5.27 |

Letter indicators (a, b, c, etc.) next to the averages refer to the so-called statistically homogeneous groups. The same letter indicator next to averages (at least one) means that there is no statistically significant difference in $p_{0.05}$ between them.

The highest dry matter and starch content directly after harvest was found in the 'Satsumo Imo' cultivar, and the lowest in 'Carmen Rubin'; whereas 'Satsumo Imo', 'Purple' and 'White Triumph' turned out to be homogeneous in terms of this characteristic. The conditions in the vegetation period also differentiated the content of dry matter and starch. The highest dry matter and starch content in tubers was recorded in 2017, with a wet summer and a dry autumn, which allowed the plants to accumulate larger amounts of these reserve materials in tubers; the lowest amount of dry matter and starch was observed in 2015; whereas in 2015 and 2016, with unfavourable humidity and temperature (wet years), the content of dry matter and starch turned out to be homogeneous (Table 11). The greatest concentration of sugars in tubers was found in the 'Carmen Rubin' cultivar, and the lowest in 'White Triumph'; whereas 'White Triumph' and 'Purple' did not manifest significant differences in terms of this characteristic. In 2017, which promoted dry matter accumulation, the plants also accumulated the most sugars in tubers. The least sugars were recorded in 2016, with a wet and cool summer and autumn; whereas there was no significant difference between total sugars in 2015 and 2017 (Table 11). Moreover, no cultivars x years interaction was found for this tuber quality characteristic.

### 3.7. Dry Matter Content in Tubers after Storage

After 6 months of storage, significant differences in dry matter content in tubers were observed. A significantly higher value of this characteristic was found in object stored at 15 °C, and significantly lower in objects stored at 5 °C (Table 12).

**Table 12.** Dry matter content in sweet potato tubers after 6 months of storage depending on storage temperature, cultivar and storage season (g kg$^{-1}$ DM).

| Experimental Factors | | Storage Seasons | | | Average |
|---|---|---|---|---|---|
| | | 2015/2016 | 2016/2017 | 2017/2018 | |
| **Storage Temperature** | 5 °C | 28.34 ± 4.65 [b] | 29.23 ± 4.66 [b] | 29.99 ± 4.79 [b] | 29.19 [b] |
| | 15 °C | 32.70 ± 6.00 [a] | 33.60 ± 6.13 [a] | 34.46 ± 5.97 [a] | 33.59 [a] |
| **Cultivars** | 'Carmen Rubin' | 22.52 ± 1.12 [b] | 23.05 ± 1.23 [b] | 23.76 ± 1.63 [b] | 23.11 [d] |
| | 'White Triumph' | 36.70 ± 2.67 [a] | 37.82 ± 2.58 [a] | 38.50 ± 2.58 [a] | 37.67 [a] |
| | 'Beauregard' | 27.33 ± 2.31 [b] | 28.65 ± 1.59 [a] | 29.68 ± 1.66 [a] | 28.56 [c] |
| | 'Satsumo Imo' | 32.23 ± 1.10 [a] | 32.78 ± 1.81 [a] | 33.65 ± 1.70 [a] | 32.89 [ab] |
| | 'Purple' | 33.83 ± 4.82 [a] | 34.78 ± 4.76 [a] | 35.53 ± 4.71 [a] | 34.71 [ab] |
| **Average** | | 30.52 | 31.42 | 32.23 | 31.39 |

Letter indicators (a, b, c, etc.) next to the averages refer to the so-called statistically homogeneous groups. The same letter indicator next to averages (at least one) means that there is no statistically significant difference in $p_{0.05}$ between them.

Genetic features of the cultivars also influenced the value of this characteristic. The cultivar with the highest dry matter content after storage was 'White Triumph', and with the lowest–'Carmen Rubin'. In season 2015/2016, the lowest amount of dry matter was recorded in 'Carmen Rubin', but 'Beauregard' was homogeneous in terms of this characteristic; similarly, cultivars homogeneous in terms of this characteristic in that storage season were 'White Triumph', 'Satsumo Imo' and 'Purple'. In the remaining storage seasons, the significantly lowest dry matter content was found in 'Carmen Rubin', whereas the other cultivars were homogeneous in terms of this characteristic (Table 12).

In the case of dry matter content in sweet potato tubers, interaction was recorded between cultivars and storage seasons. Only the 'White Triumph' cultivar responded with a significant increase in dry matter content to storage conditions in seasons 2016/2017 and 2017/2018, as compared to season 2015/2016 (Figure 2).

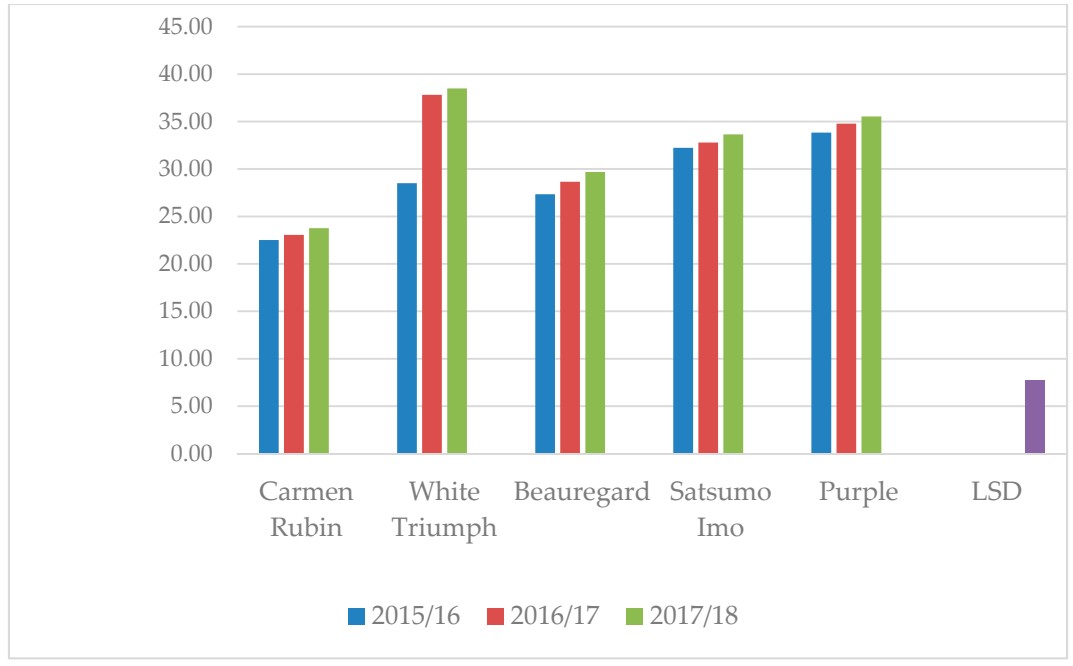

**Figure 2.** Influence of cultivar and storage season on dry matter content in tubers (% DM).

### 3.8. Starch Content in Tubers after Storage

After 6 months of storage, starch content was significantly differentiated by all experimental factors (Table 13). Tuber storage at 15 °C contributed to a significant decrease of starch content, as compared to tuber storage at 5 °C.

**Table 13.** Starch content in sweet potato tubers after 6 months of storage depending on storage temperature, cultivar and storage season (g kg$^{-1}$ FM).

| Experimental Factors | | Storage Seasons | | | Average |
| --- | --- | --- | --- | --- | --- |
| | | 2015/2016 | 2016/2017 | 2017/2018 | |
| Storage Temperature | 5 °C | 17.20 ± 2.22 [a] | 17.16 ± 2.28 [a] | 17.11 ± 2.28 [a] | 17.16 [a] |
| | 15 °C | 14.64 ± 2.02 [b] | 14.58 ± 1.80 [b] | 14.77 ± 1.86 [b] | 14.68 [b] |
| Cultivars | 'Carmen Rubin' | 13.43 ± 1.09 [a] | 12.93 ± 0.92 [a] | 13.07 ± 1.03 [a] | 13.05 [c] |
| | 'White Triumph' | 18.21 ± 1.33 [a] | 18.03 ± 0.98 [a] | 18.20 ± 0.99 [a] | 18.03 [a] |
| | 'Beauregard' | 15.71 ± 1.19 [a] | 16.08 ± 1.06 [a] | 15.33 ± 1.18 [a] | 15.51 [b] |
| | 'Satsumo Imo' | 19.16 ± 1.63 [a] | 17.64 ± 2.64 [a] | 18.09 ± 2.15 [a] | 18.21 [a] |
| | 'Purple' | 14.76 ± 1.76 [a] | 14.68 ± 1.50 [a] | 15.03 ± 1.11 [a] | 14.84 [b] |
| Average | | 15.92 | 15.87 | 15.94 | 15.91 |

Letter indicators (a, b, c, etc.) next to the averages refer to the so-called statistically homogeneous groups. The same letter indicator next to averages (at least one) means that there is no statistically significant difference in $p_{0.05}$ between them.

Genetic features of the cultivars had a significant influence on the accumulation of starch in sweet potato tubers. The most starch remained in the 'Satsumo Imo' cultivar, and the least in 'Carmen Rubin'. This characteristic turned out to be homogeneous for the 'Purple' and 'Beauregard', as well as 'Satsumo Imo' and 'White Triumph' cultivars. The conditions in the vegetation period did not differentiate starch content in tubers. No cultivar x year interaction was observed (Table 13).

### 3.9. Sugar Content after Storage

Total sugars in tubers after 6 months of storage was differentiated and ranged from 4.12 to 9.64 g kg$^{-1}$ of tuber fresh matter (FM) (Table 14).

**Table 14.** Total sugars in sweet potato tubers after 6 months of storage depending on storage temperature, cultivar and storage season (g kg$^{-1}$ FM).

| Experimental Factors | | Storage Seasons | | | Average |
| --- | --- | --- | --- | --- | --- |
| | | 2015/2016 | 2016/2017 | 2017/2018 | |
| Storage Temperature | 5 °C | 5.75 ± 1.81 [b] | 5.88 ± 1.88 [b] | 6.28 ± 1.77 [b] | 5.97 [b] |
| | 15 °C | 6.85 ± 1.93 [a] | 7.16 ± 2.15 [a] | 8.17 ± 1.77 [a] | 7.39 [a] |
| Cultivars | 'Carmen Rubin' | 9.38 ± 0.65 [a] | 9.27 ± 0.77 [a] | 9.64 ± 0.69 [a] | 9.43 [a] |
| | 'White Triumph' | 4.28 ± 0.34 [b] | 4.12 ± 0.63 [b] | 5.30 ± 1.29 [b] | 4.57 [c] |
| | 'Beauregard' | 6.81 ± 0.79 [a] | 7.44 ± 1.20 [a] | 8.00 ± 1.69 [a] | 7.42 [b] |
| | 'Satsumo Imo' | 6.23 ± 0.64 [a] | 7.23 ± 0.62 [a] | 7.82 ± 0.98 [a] | 7.09 [b] |
| | 'Purple' | 4.80 ± 0.71 [b] | 4.53 ± 0.38 [b] | 5.37 ± 0.55 [b] | 4.90 [c] |
| Average | | 6.30 | 6.52 | 7.23 | 6.68 |

Letter indicators (a, b, c, etc.) next to the averages refer to the so-called statistically homogeneous groups. The same letter indicator next to averages (at least one) means that there is no statistically significant difference in $p_{0.05}$ between them.

During a long storage period, a significant increase of total sugars was recorded at 15 °C as compared to storage at 5 °C. +The most sugars were accumulated in the 'Carmen Rubin' tubers, and less than a half of that was found in the 'White Triumph' cultivar; whereas the 'White Triumph' and 'Purple', as well as 'Beauregard' and 'Satsumo Imo' cultivars turned out to be homogeneous in terms of this characteristic (Table 14). The 'White Triumph' and 'Purple' cultivars can be considered CIS-tolerant, whereas 'Carmen Rubin', 'Beauregard' and 'Satsumo Imo'–CIS-sensitive.

Weather conditions in the research years also influenced sugar accumulation in the stored tubers. The greatest concentration of sugars was recorded in the season following extreme drought in September 2017, and the lowest–after wet and cool vegetation in 2015. The cultivars under study responded differently to vegetation conditions. In all storage seasons, the 'White Triumph' and 'Purple' cultivars turned out to have a significantly lower sugar content than the other cultivars, all of which belonged to the same homogeneous group (Table 14).

A significant increase in total sugars was recorded in the 'Carmen Rubin' and 'Satsumo Imo' cultivars in storage season 2017/2018 as compared to seasons 2015/2016 and 2016/2017 (Figure 3).

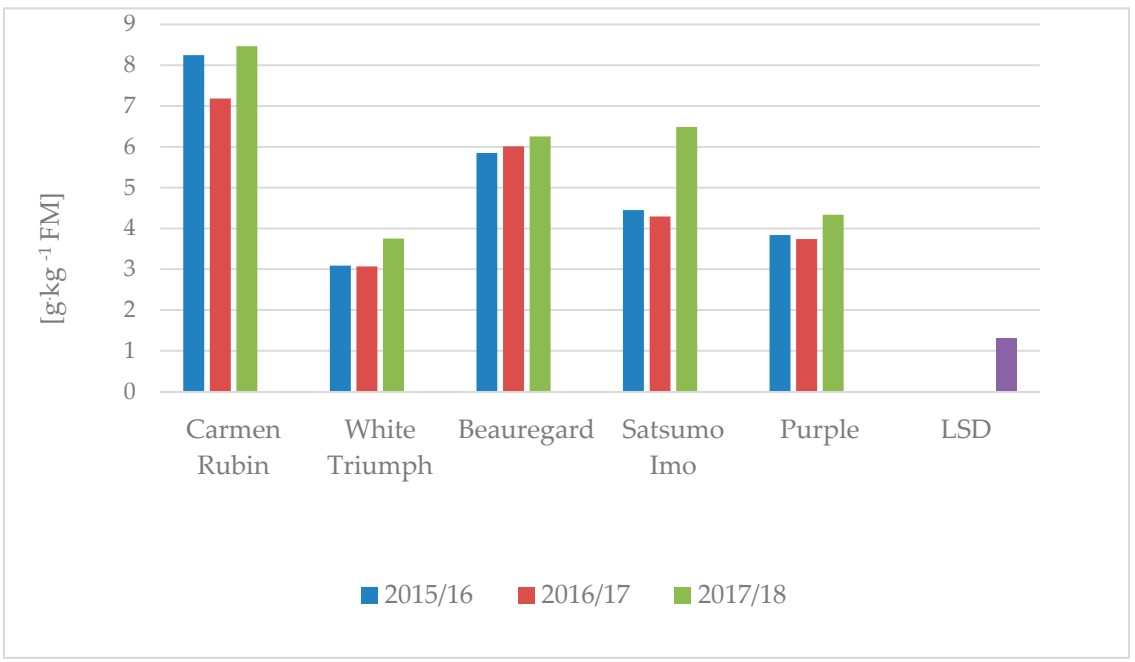

**Figure 3.** Influence of cultivars and storage seasons on total sugars.

## 4. Discussion

### 4.1. The Influence of Edaphic Factors on Tuber Weight Losses and Waste During Storage

In the conducted research, tubers stored at the higher temperature of 15 °C sprouted earlier than tubers stored at a temperature that was 10 °C lower. Edmunds et al. [32], proved that germination at a higher temperature is a natural process, but results in quick sweet potato tuber weight loss, which leads to decreased firmness and consumer and economic value. It was proven that long-term storage of *Ipomoea batatas* tubers requires the use of sprout inhibitors, whereas the stimulation of this process is required for the production of planting material, so it is recommended storage practice to use inhibitors and growth stimulants (e.g., gibberellin, ethylene) [33–37]. Unlike potato tubers, sweet potato tubers do not undergo dormancy and can sprout at any time after harvest, provided the conditions are favourable, i.e., suitable temperature (12–15 °C) and relative humidity (80–95%) [38]. Tuber storage at higher temperatures promotes sprout growth. According to Woolfe [27], Edmunds et al. [32], sprouted tubers are characterised by higher respiration rate, which leads to higher weight loss. Sonnewald and Sonnewald [39] noted that sucrose availability is a prerequisite for bud break. In the absence of this

disaccharide, no bud break occurs. Thus, sucrose is likely to serve as nutrient and signal molecule at the same time. It most likely involves trehalose-6-phosphate and SnRK1 signalling networks. Sprouting leads to major quality losses of stored potato tubers. Therefore, control of tuber sprouting is a major objective in sweet potato breeding. In New Zealand, sprouting is considered one of the key criteria to deem sweet potato tubers marketable [40]. Losses due to tuber sprouting also have an economic and social aspect, because such tubers cannot be sold at a higher price. The results of own research indicate that sprouting-based changes were limited and could have been prevented by spraying with natural growth inhibitors, e.g., cumin oil solution. This should be the direction of future studies on delaying tuber sprouting.

According to Dandago and Gungula [10], during storage, the indicators determining natural losses are physiological processes (transpiration and respiration) and the progress of thermal and humidity conditions in storage. Shuzbusha et al. [41], and Grace et al. [8] proved that the higher the temperature in storage, the more intensive the transpiration, which results in higher natural losses. They believe that such growth is ostensible and related to water loss. According to Padda and Picha [42], storage temperatures below 12 °C can cause losses due to cooling, leading to weight losses, deteriorated flavour and tuber rotting.

The length of the storage period depends on the cultivar. It is based on various rates of respiration, transpiration, sprouting ability and pathogen susceptibility [15,17,32,43]. Edmunds et al. [32] proved that sweet potato cultivars differ in terms of efficiency of wound healing during preparation for storage. The ability of *Ipomoea batatas* to regenerate damages that occurred during harvest and transport is necessary to prevent excessive water loss and pathogen penetration through wounds during storage. Hence, sweet potato tubers demonstrating a stronger suberization tendency can be stored for a longer time [44]. Rykaczewska [45], demonstrated that cultivar differences in terms of tuber losses and waste during storage result from their physiological and chronological age, because plant stress caused by high ambient temperature during vegetation–an agricultural problem in many regions–results in a number of morphological, anatomical, physiological and biochemical changes in plants, which have a direct effect on their growth and development. Such changes can lead to a significant decrease in crop size and tuber quality. Negative effects of high temperature can be mitigated by introducing improved thermal control to cultivation. Post-harvest maturation of *Ipomoea batatas* tubers results in corky skin, which a barrier to pathogens and reduces dehydration of scraped skin, contributing to decreased tuber weight loss during storage [38]. A number of authors Edmunds et al. [32], Tomlins et al. [17], Chakraborty et al. [15], believe that tubers that were not subject to post-harvest maturation do not respond well to storage, whereas tubers subject to maturation at 13°–18 °C with 85–95% RH can be stored for a period of up to 12 months. According to Dandago and Gungula [10], post-harvest maturation should not exceed 5–7 days, because higher temperature at this stage results in sprouting, which in turn can cause tuber weight loss.

During storage, tubers lose firmness (turgor pressure) and undergo biochemical and physiological changes leading to natural losses not only due to germination, but alto respiration and transpiration, as well as tuber rotting waste. From the economic and technological perspective, transpiration is the most important among these processes [15,17,32]. Changes resulting from excessive transpiration of *Ipomoea batatas* tubers affect a number of characteristics determining their consumer and processing value, which is important in their use for direct consumption and processing into enriched products [17,32]. In the conducted research, transpiration and respiration losses were relatively high, amounting to approx. 5.6%, and depending on cultivar and vegetation conditions–from 4.0 to 7.6%. According to a number of authors Dandago and Gungula [10], Cheema et al. [34], Ali et al. [46], Nabubuya et al. [47], Aswathy et al. [12], transpiration rate depends on many factors, e.g., cultivar, physiological ripeness of tubers (skin suberization) and storage conditions. Storage losses are determined by the following factors: temperature, humidity and ventilation rate [48]. Thus, the conditions of the plant growing season had the greatest impact on the amount of losses.

The greatest losses during 6 months of storage, in own research, resulted from tuber rotting. Storage temperature influenced this characteristic the most. Tuber storage at 15 °C reduced it over tenfold as compared to storage at 5 °C. Agu et al. [49], reported that the main pathogens in stored tubers are moulds, which cause huge post-harvest waste in crops, as well as economic losses for sweet potato growers. The identified tuber spoilage moulds included: *Aspergillus fumigatus, Aspergillus niger* and *Rhizopus stolonifer*. These isolates were subjected to pathogenicity tests using fresh healthy tubers in order to confirm their ability to elicit same spoilage symptoms in healthy tubers. Dry rot was recorded in all the tubers on a low level. A value of $98.29 \pm 0.35$ with $p > 0.1$ was recorded for the percentage rot severity.

Another factor that had a strong influence on tuber weight loss during storage consisted of genetic features of the cultivars under study. Agu et al. [49], but also Yildirim et al. [28] confirm this relation. Grudzińska and Barbaś [50], referred to resistance to storage diseases and storage waste and losses as storage stability. Agu et al. [49] points out that an increase in losses by even one percent entails significant financial losses not only for the producer, but also in terms of quality for the consumer and industry. Therefore, the choice of variety is very important because, under the same conditions, much more can be obtained from well-kept cultivars commercial goods of higher quality with lower financial outlays.

Weather conditions in the own research, in the experiment years also shaped the volume of tuber rotting waste. Higher waste was recorded in years with high rainfall during vegetation. This has been confirmed by research by Czerko and Grudzińska [51], Adu-Kwarteng et al. [48], Grudzińska and Mańkowski [52]. Earlier studies confirmed greater convergence natural losses with the level of rainfall than air temperature and greater losses were found in the years with a lack of rainfall during the growing season, and smaller in the years with rainfall above the multi-year average [10,48].

*4.2. The Influence of Tuber Chemical Composition on naTural Weight Losses and Waste*

The conducted research determined the content of basic components of dry matter after harvest and after storage. Significant differences were recorded involving increased dry matter content and total sugars, and at the same time, decreased starch content in tubers after storage. It is a known fact that this results from physiological and biochemical changes in tubers during storage, but it is also determined by endogenous factors. Yamdeu et al. [18], studied carbohydrate metabolic changes in potato tubers stored at 15 °C and 4 °C for 150 days in order to understand the development of cold-induced sweetening (CIS). They proved that low-temperature storage negligibly influenced starch and maltose contents of the tubers but induced a significant increase of reducing sugars, total soluble sugars, fructose, glucose and hexoses: sucrose ratio, and a decrease of sucrose content was noticeable at 4 °C. The authors found a strong positive correlation between reducing sugars and total soluble sugars, and between fructose and glucose. The activity of β-amylase was considerably increased by storage at low temperature, and it weakly correlated with starch content. Amjad et al. [13], proved that no maltose accumulation with increased β-amylase activity also leads to reduced starch content, whereas the activity of acid invertase drastically increased at low temperature and at the same time, the ratio of reducing sugars–glucose, fructose and hexose–to sucrose changed. The key enzyme involved in the breakdown of starch is acid invertase, whereas β-amylase is not a key enzyme in starch conversion and significant maltase activity is possible in potato tubers. During low temperature storage of tubers, starch breaks down while sucrose is formed by UDP-glucose pyrophosphorylase and sucrose-phosphate synthase. Sucrose is then hydrolysed to RS by soluble acid invertase enzyme, a key enzyme involved in the hydrolysis of sucrose to glucose and fructose. Invertase facilitates the hydrolytic breakdown of sucrose into hexose monomers to fulfil the plant physiological requirements of carbohydrate transport, stress response and sugar signalling [13]. These observations paved the way for biotechnology works necessary to develop new sweet potato cultivars that would handle this post-harvest problem.

The conducted research also enabled a division of the sweet potato cultivars under study into low and high-sugar content groups, which will help identify cultivars suitable for processing directly after harvest or short-term storage, as well as identify forms with high starch content suitable for starch extraction. This will provide the processing industry with basic information on cultivars with high starch yield, which can be stored in a cool store with no dramatic sugar increase. This can be also useful to growers looking for CIS-resistant genes in sweet potato cultivars adapted to the Central-Eastern European climate. These observations, however, should be furthered, so that during CIS development in these cultivars, biotechnological works can step in and create new sweet potato cultivars that will be resistant to the issue of post-harvest sugar content increase in tubers. This, however, requires a more detailed discussion. It is a known fact that the development and growth of tubers is strongly influenced by environmental and endogenous factors. In the case of potato (*Solanum tuberosum*), tuber induction conditions are well defined. But in the case of sweet potato, low air temperature (14–22 °C), high sucrose content and appropriate soil humidity can facilitate tuber formation, starch synthesis and storage root weight increase, despite the fact that the threshold value for each of these factors is not clearly defined. In favourable conditions, in case of earlier high-yield sweet potato cultivars, storage tubers are developed already in week three after planting. Nevertheless, the formation and growth of storage roots in long-vegetation cultivars can be delayed [53]. But expression of genes involved in the early stage of storage root formation and their amount require further research. When sweet potato shoots start root formation, some of them develop into storage roots, whereas others become lignified and only grow to be pencil-thick. This usually occurs under stress, such as water excess (water stress) or drought stress [52]. The effect of biosynthesis gene expression on biochemical changes in roots during sweet potato storage requires further study.

All experimental factors influenced the chemical composition of tubers evaluated after storage. Shuzbusha et al. [41], and Grace et al. [8], proved that the higher the storage temperature, the more intensive the transpiration, which results in higher dry matter content in tubers. Our own research corroborates this thesis. According to Dandago and Gungula [10], during storage, the indicators determining dry matter content changes are physiological processes (transpiration and respiration) and the progress of temperature and humidity in storage. In our own research, dry matter content was modified by significant genetic features of the cultivars. The same results were also arrived at by Ali et al. [46], Gwandu et al. [54], Kathabwalika et al. [55], who achieved similarly large differences of dry matter content between the cultivars (from 13.4% to 41.1%). After 6 months of sweet potato tuber storage at 5 °C and 15 °C, a change was also observed in the content of starch, which is the fundamental carbohydrate in sweet potato tubers. According to a number of authors [6,56–58] its content is directly related to dry matter content. Dandago and Gungula [10], noted that starch content in sweet potato tubers decreases over time during storage. Zhanga et al. [59], also observed a decrease in starch content during *Ipomoea batatas* tuber storage; however, it differed depending on genotype. Nabubuya et al. [48,60], demonstrated that starch content decrease in sweet potato tubers during storage is a result of enzyme activity, in particular of amylases. Their activity in sweet potato tubers increases during storage and plays an important role in decreasing starch content during storage or germination [60].

The conducted research also showed that during storage, total sugars also change in tubers. Nabubuya et al. [60], demonstrated that long storage of sweet potato results in increased total sugars in tubers. The carbohydrate fraction is changed by amylase enzymes, which hydrolyse glycosidic linkages in starch granules leading to increased amount of monosaccharides [38,49]. Katayama et al. [61] also proved that during sweet potato tuber storage, total sugars increase, resulting in deteriorated culinary properties.

Grudzińska and Barbaś [50], corroborate our assumptions that in the case of tubers prone to internal black spot after mechanical damage during transportation and preparation for storage, natural losses due to storage at higher temperatures can be a subjective method of evaluating sweet

potato susceptibility to internal black spot. However, this observation requires further research and substantiation of this theory.

## 5. Conclusions

Natural losses and waste as well as sweet potato tuber quality were mostly determined by storage temperature. Sweet potato tuber storage at 15 °C contributed to a significant reduction in waste and weight losses due to germination, transpiration, respiration and rotting, but also to an increase in dry matter content and total sugars and a decrease in starch content in tubers, as compared to storage at 5 °C. Regardless of temperature, storage stability was shaped by genetic features of the cultivars under study. Cultivars demonstrating long-term storage stability were 'Purple and 'Satsumo Imo', whereas the least storage stability was observed in 'Carmen Rubin'. Cultivars that maintained the best quality after 6 months of storage were 'Purple' and 'White Triumph'. The remaining cultivars should be stored for a shorter time so that they can preserve their good properties. For the purpose of identifying cultivars suitable for processing directly after harvest or short-term storage and identifying cultivars with high starch content suitable for starch extraction (e.g., 'White Triumph' and 'Satsumo Imo'), the cultivars under study were divided into low and high sugar content groups. A different response of the 'Satsumo Imo', 'Carmen Rubin' and White Triumph' cultivars to the conditions in the two experimental years, i.e., sugar content increase and dry matter accumulation in tubers, resulted from their genetic features. The experiment results can contribute to developing storage technology of sweet potato tubers cultivated in Central Europe.

**Author Contributions:** Conceptualization, B.K.-M. and B.S.; methodology, B.K.-M.; B.S. and B.K.; software, B.K.-M.; validation, B.K.-M.; B.S. and B.K.; formal analysis, B.K.-M.; investigation, B.K.-M.; resources, B.K.-M. and B.S.; data curation, B.K.-M.; writing—original draft preparation, B.K.-M. and B.S.; writing—review and editing, B.K.-M.; B.S.; H.D. and E.J.; visualization, B.K.-M.;B.S.; H.D. and E.J.; supervision, B.K.-M.; project administration, B.K.-M.; funding acquisition, B.K.-M. All authors have read and agreed to the published version of the manuscript.

**Funding:** This research was funded by the Scholarship Fund of Stanisława Pigoń for students and employees of the State Higher Vocational School in Krosno with the number PDR.SP.0041.6.2020.

**Conflicts of Interest:** The authors declare no conflict of interest.

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
