# Peer review of "The Effects of Temperature on the Quality and Storage Stalibity of Sweet Potato (Ipomoea batatas L. [Lam]) Grown in Central Europe"

_agronomy, doi:10.3390/agronomy10111665_

Round 1
Reviewer 1 Report
Please make the corrections marked in the text.

Author Response
Manuscript ID: agronomy-978730
Title: The effects of temperature on the quality and storage stability of sweet potato (Ipomoea batatas L. [Lam]) grown in Central Europe
Dear Editor,
We are very grateful for the insightful review of our manuscript. The reviewers' remarks and suggestions, which have been implemented into the text, will significantly increase the scientific value of it. The text has been improved according to the reviewers' remarks, and below we have enclosed comments on the revised version. We hope that the manuscript in its present form meets the requirements of the Agronomy.
Yours Sincerely,
Dr inż. Barbara Krochmal-Marczak
Carpathian State College in Krosno
Department of Plant Production and Food Safety,
38-400 Krosno, Poland;
barbara.marczak@kpu.krosno.pl
Response to Reviewer 1 Comments
Thank you for the remarks and suggestions of Reviewer 1. The manuscript has been corrected according to the remarks and suggestions of Reviewer 1.
Point 1: In line 162, the error “700°C”
Response 1: In line 162, the error “700°C” was corrected to “70°C”.
Point 2: In line 495, the error “Nabubuga et al. [47]”
Response 2: In line 495, the error “Nabubuga et al. [47]” was corrected to “Nabubuja et. al. [47]”.
Point 3: In line 516, the error “Agu et al. [48]”
Response 3: In line 516, the error “Agu et al. [48]” was corrected to “Adu-Kwarteng et. al. [48]”.
Point 4: The Reviewer’s remark to “Standardize the drawing description” in Figure 1 and Figure 2
Response 4: The Reviewer’s remark to “Standardize the drawing description” in Figure 1 and Figure 2 was implemented and the errors corrected.
Point 5: Reviewer 1’s remark “, Instead of a general summary, please present the results in 3-4 synthetic conclusions.”
Response 5: Due to the requirements of the Agronomy journal, the conclusions remain in a descriptive form. We would like to ask the Reviewer 1 to agree to such form.

Reviewer 2 Report
The manuscript “The effects of temperature on the quality and storage stability of sweet potato (Ipomoea batatas L. [Lam]) grown in Central Europe” seems to be an interesting topic. The authors tried to explain the effect of storage temperature on quality changes of some sweet potato cultivars. Although it is very difficult to correlate the quality and storage stability with some environmental parameters like temperature, rainfall etc. They also tried to draw a relationship between storage environment and culinary properties of sweet potato cultivars. More interestingly they used 3 years relevant data which I appreciate most. However, some queries need to address to improve the manuscript. I suggest including some information for which I recommend for minor revision.
Results and discussions:
Line 225-226: Please write the results exactly showing in the table (natural sprouting losses, natural transpiration and respiration losses).
Table 9: Natural transpiration and respiration losses in 2017/2018 season doesn’t show statistical accuracy. Please check letter indicators among the cultivars (specially between White triumph and Satsumo Imo).
Line 312-315: Author mentioned “The cultivar most prone to tuber rotting was ‘Carmen Rubin’, and the least – ‘Purple’, whereas ‘Carmen Rubin’ and ‘White Triumph’, as well as ‘Beauregard’, ‘Satsumo Imo’ and ‘Purple’ turned out to be homogeneous in terms of this characteristic”. However, table 10 showing purple is homogeneous in 2017/2018 season only which is mentioned in last part of results (line 326-327). Please fixed this issue.
Line 367-368: What do you mean by vegetation conditions? The cultivar Beauregard is already showing significant differences regarding dry matter content in different years (Table 12). So, how would you say “vegetation conditions in the research years did not differentiate dry matter content in tubers” any supporting data regarding this information?
Line 390: Author mentioned “Genetic features of the cultivars had a significant influence on the accumulation of starch in sweet potato tubers” but I did not find any significant differences among the cultivars and storage years. Did you mean average starch accumulations?
Line 408-409: Which cultivars (CIS-tolerant) are you indicating as latter two? Please check whether the word ‘latter’ is appropriate or not.
Line 413-417: As all the cultivars gave similar results in three consecutive seasons. I think no need to write two sentences separately, rather include 2015/2016 season along with others two in 415 line.
Line 421-422: “A significant increase of sugar content in season 2017/2018” compare to which season/s?
Discussion
The discussion part seems like full of generalized statement specially in 4.1 section. It would be better to add more in-depth analysis and provide specific information regarding the data obtained from the study. It would be more interesting if possible, to analyze how environmental parameters effect on post-harvest quality and that should focus more precisely by using some relevance references.
Author Response
Manuscript ID: agronomy-978730
Title: The effects of temperature on the quality and storage stability of sweet potato (Ipomoea batatas L. [Lam]) grown in Central Europe
Dear Editor,
We are very grateful for the insightful review of our manuscript. The reviewers' remarks and suggestions, which have been implemented into the text, will significantly increase the scientific value of it. The text has been improved according to the reviewers' remarks, and below we have enclosed comments on the revised version. We hope that the manuscript in its present form meets the requirements of the Agronomy.
Yours Sincerely,
Dr inż. Barbara Krochmal-Marczak
Carpathian State College in Krosno
Department of Plant Production and Food Safety,
38-400 Krosno, Poland;
barbara.marczak@kpu.krosno.pl
Response to Reviewer 2 Comments
Thank you for the remarks and suggestions of Reviewer 2. The manuscript has been corrected according to the remarks and suggestions of Reviewer 2.
Point 1: Line 225-226: Please write the results exactly showing in the table (natural sprouting losses, natural transpiration and respiration losses).
Response 1: We would like to thank Reviewer 2 for this important remark. Corrections were made and data reflecting Table 5 results was added.
Average natural tuber losses and waste during 6 months of storage were as follows: natural sprouting losses amounted to just 0.41%, natural transpiration and respiration losses –5.61%, rotting – 26.56% waste of tuber weight. In total, tuber weight losses and waste during 6 months of storage amounted to 32.59%, depending on temperature – 2.17–53.01%, depending on cultivar – 27.68–40.15% and depending on storage season – 31.98–33.29% (Table 5).
Point 2: Table 9: Natural transpiration and respiration losses in 2017/2018 season doesn’t show statistical accuracy. Please check letter indicators among the cultivars (specially between White Triumph and Satsumo Imo).
Response 2: We would like to thank Reviewer 2 for this important remark. Errors in White Triumph and Satsumo Imo data in the 2017/2018 storage season were corrected.
Point 3: The cultivar most prone to tuber rotting was ‘Carmen Rubin’, and the least – ‘Purple’, whereas ‘Carmen Rubin’ and ‘White Triumph’, as well as ‘Beauregard’, ‘Satsumo Imo’ and ‘Purple’ turned out to be homogeneous in terms of this characteristic.
Response 3: The sentence was corrected to:
The cultivar most prone to tuber rotting was ‘Carmen Rubin’, and the least – ‘Purple’, whereas ‘Beauregard’ and ‘Satsumo Imo’ turned out to be homogeneous in terms of this characteristic.
Point 4: Line 312-315: Author mentioned “The cultivar most prone to tuber rotting was ‘Carmen Rubin’, and the least – ‘Purple’, whereas ‘Carmen Rubin’ and ‘White Triumph’, as well as ‘Beauregard’, ‘Satsumo Imo’ and ‘Purple’ turned out to be homogeneous in terms of this characteristic”. However, table 10 showing purple is homogeneous in 2017/2018 season only which is mentioned in last part of results (line 326-327). Please fixed this issue.
Response 4: We would like to thank Reviewer 2 for the remark. The text was corrected expanded.
The greatest storage waste during 6 months of storage was due to tuber rotting. The factor differentiating this characteristic the most was storage temperature. Tubers stored at 5°C were several times more prone to rotting that tubers stored at 15°C. Tubers stored at lower temperatures were more susceptible to rot despite lower water loss. The cultivar most prone to tuber rotting was ‘Carmen Rubin’, and the least – ‘Purple’, whereas ‘Carmen Rubin’ and ‘White Triumph’, as well as ‘Beauregard’, ‘Satsumo Imo’ and ‘Purple’ turned out to be homogeneous in terms of this characteristic. The greatest rotting waste in tuber storage occurred in the 2015/2016 storage season, and the least tubers affected by fungal and bacterial pathogens were recorded in the 2017/2018 season In storage seasons 2015/2016 and 2016/2017, the cultivar most resistant to tuber rotting was Purple, whereas the remaining cultivars turned out to be homogeneous in terms of this characteristic. In season 2017/2018, all cultivars were homogeneous in terms of this characteristic (Table 10).
Point 5: Line 367-368: What do you mean by vegetation conditions? The cultivar Beauregard is already showing significant differences regarding dry matter content in different years (Table 12). So, how would you say “vegetation conditions in the research years did not differentiate dry matter content in tubers” any supporting data regarding this information?
Response 5: We would like to thank Reviewer 2 for the remark. The text was corrected and explanation was added.
Genetic features of the cultivars also influenced the value of this characteristic. The cultivar with the highest dry matter content after storage was ‘White Triumph’, and with the lowest – ‘Carmen Rubin’. In season 2015/2016, the lowest amount of dry matter was recorded in ‘Carmen Rubin’, but ‘Beauregard’ was homogeneous in terms of this characteristic; similarly, cultivars homogeneous in terms of this characteristic in that storage season were ‘White Triumph’, ‘Satsumo Imo’ and ‘Purple’. In the remaining storage seasons, the significantly lowest dry matter content was found in ‘Carmen Rubin’, whereas the other cultivars were homogeneous in terms of this characteristic.
Point 6: Line 390: Author mentioned “Genetic features of the cultivars had a significant influence on the accumulation of starch in sweet potato tubers” but I did not find any significant differences among the cultivars and storage years. Did you mean average starch accumulations?
Response 6: Thank you for the important remark. By writing
“Genetic features of the cultivars had a significant influence on the accumulation of starch in sweet potato tubers”
we meant average starch content.
Point 7: Line 408-409: Which cultivars (CIS-tolerant) are you indicating as latter two? Please check whether the word ‘latter’ is appropriate or not.
Response 7: Thank you for the remark of Reviewer 2. The text was corrected from
The latter two can be considered CIS-tolerant, whereas ‘Carmen Rubin’, ‘Beauregard’ and ‘Satsumo Imo’ – CIS-sensitive.
to:
The ‘White Triumph’ and ‘Purple’ cultivars can be considered CIS-tolerant, whereas ‘Carmen Rubin’, ‘Beauregard’ and ‘Satsumo Imo’ – CIS-sensitive.
Point 8: Line 413-417: As all the cultivars gave similar results in three consecutive seasons. I think no need to write two sentences separately, rather include 2015/2016 season along with others two in 415 line.
Response 8: The remark was implemented:
Weather conditions in the research years also influenced sugar accumulation in the stored tubers. The greatest concentration of sugars was recorded in the season following extreme drought in September 2017, and the lowest – after wet and cool vegetation in 2015. The cultivars under study responded differently to vegetation conditions. In the 2015/2016 season, the ‘Carmen Rubin’, Beauregard’ and ‘Satsumo Imo’, as well as ‘White Triumph’ and ‘Purple’ belonged to the same homogeneous groups. In the 2016/2017 and 2017/2018 seasons, a significantly lower sugar content after storage was recorded in the ‘White Triumph’ and ‘Purple’ cultivars, whereas the others belonged to a group with higher sugar content in tubers (Table 14). In all storage seasons, the ‘White Triumph’ and ‘Purple’ cultivars turned out to have a significantly lower sugar content than the other cultivars, all of which belonged to the same homogeneous group (Table 14).
Point 9: Line 421-422: “A significant increase of sugar content in season 2017/2018” compare to which season/s?
Response 9: Thank you for the remark of Reviewer 2. The text was corrected from
The sweet potato cultivars under study responded in different ways to storage conditions. Only the ‘Carmen Rubin’ and ‘Satsumo Imo’ cultivars responded with a significant increase of sugar content in season 2017/2018 (Figure 3).
to:
A significant increase in total sugars was recorded in the ‘Carmen Rubin’ and ‘Satsumo Imo’ cultivars in storage season 2017/2018 as compared to seasons 2015/2016 and 2016/2017.
Point 10: Discussion
The discussion part seems like full of generalized statement specially in 4.1 section. It would be better to add more in-depth analysis and provide specific information regarding the data obtained from the study. It would be more interesting if possible, to analyze how environmental parameters effect on post-harvest quality and that should focus more precisely by using some relevance references.
Response 10: Thank you for the Reviewer’s remarks and suggestions. The Discussion was corrected and expanded, in particular in point 4.1. All additions to the text were saved in the Track Changes mode.
